# Assessment of Saudi Mothers’ Attitudes towards Their Children’s Pain and Its Management

**DOI:** 10.3390/ijerph18010348

**Published:** 2021-01-05

**Authors:** Sultan M. Alghadeer, Syed Wajid, Salmeen D. Babelghaith, Mohamed N. Al-Arifi

**Affiliations:** Clinical Pharmacy Department, College of Pharmacy, King Saud University, Riyadh 11451, Saudi Arabia; salghadeer@ksu.edu.sa (S.M.A.); sbabelghaith@ksu.edu.sa (S.D.B.); malarifi@ksu.edu.sa (M.N.A.-A.)

**Keywords:** pain, analgesics, antibiotics, pediatrics, Saudi mother, Saudi Arabia

## Abstract

Background and objective: Pain is a bothersome issue that is common among newborns and children of all ages. Pain can be managed using various pharmacological and/or non-pharmacological strategies, which can be delivered by healthcare providers or parents. The aim of this study is to assess the attitude of mothers toward their children’s pain and its management. Methods and materials: A descriptive cross-sectional web-based study was conducted using a developed self-reported questionnaire, from March 2018 to April 2018. Participants involved were Arabic-speaking mothers of children aged between 0 and 12. The data collected included pharmacological and non-pharmacological treatment methodologies utilized to treat pain and the mothers’ attitude towards pain management. Statistical Package for Social Sciences, version 25 was applied to analyze the data, and descriptive statistics were performed. Results: As per the results of this study, the most common site of children’s pain as reported by the mothers was mouth/throat (211; 52.9%), abdomen (199; 49.9%), followed by head (58; 14.5%), and finally, ears (69; 17.3%). The frequency of children’s pain, as stated by the mothers, was less than once a month (196; 49.1%), once in a month (137; 34.3%), and once in a week (48; 12%). The non-pharmacological methods used by mothers at home for the management of their children’s pain were as follows: letting them take rest or sleep (250; 62.6%), feeding them with fluids (228; 57.1%), applying cold packs (161; 40.4%), providing massage therapy (147; 36.8%), using warm packs (141; 35.3%), and taking them to play (119; 29.8%). Conclusion: The misconceptions about pediatric pain management among Saudi mothers that can affect the children’s quality of life are quite noticeable. Implementing educational and awareness programs about the management of child pain could play a major role in making the parents understand the impacts of their misconceptions.

## 1. Background

Pain is a bothersome issue that is common among newborns and children of all ages. It accounts for more than 30% of the cases attended by pediatricians [1,2,3]. Children are more defenseless against pain than adults. They may encounter pain because of some ailment, incapacity, injury, or minor wounds, such as cuts or knocks [3,4]. For the most part, pediatric pain comes along with a disease or damage, and so there is a developing worry about it possibly causing unfavorable long-term impacts on the cerebrum. During hospitalization, children may endure continued pain due to inappropriate pain management, absence of proper paternal involvement, or a lack of mutual confidence between healthcare providers and children [5].

Management of pain involves the use of various pharmacological and/or non-pharmacological strategies, which can be delivered by healthcare providers or parents. Several studies regarding practices utilized by healthcare providers in the management of children’s pain have been identified. In recent years, however, the attention of research studies has shifted towards home management of children’s pain by their mothers. This is because it is increasingly being noticed that a greater part of children’s pain is treated by their mothers at home [6]. Although in most cases, mothers solely take care of their children’s pain, their knowledge of and attitudes toward pharmacological management of their children’s pain has been reported to be low [7,8]. Some mothers also resort to home treatments as they believe in myths about the addictive and adverse effects of medication, and thus develop negative attitudes towards analgesics [8,9].

Parents’ attitudes are essential in the health training and advancement of their children. Therefore, it is imperative to investigate parents’ attitudes towards their children’s pain and its treatment [10]. Parents’ unwillingness in allowing their children to take drugs could be due to their inadequate knowledge about pain and analgesics, as well as the benefits and safety of analgesics [11]. Even the mothers’ approaches to pain relief and their attitudes towards their children’s postoperative pain were found to be unsatisfactory [12]. Considering the importance of a mother’s role in pediatric pain management and the scarcity of information regarding their attitudes towards pain management and their appropriate use of analgesics, this study aims to assess the attitude of mothers toward their children’s pain and its management.

## 2. Methods

### 2.1. Study Population and Design

A descriptive, cross-sectional study was conducted by using a developed self-reported questionnaire in Saudi Arabia over a period of two months, between March 2018 and April 2018. Participants involved in the study were Arabic-speaking, Saudi Arabian mothers, who have children aged between 0 and 12 years. The research data was collected through online questionnaires circulated on social media platforms such as Twitter, WhatsApp, and Facebook. This online survey questionnaire consisted of three sections, containing a total of 29 binary and multiple-choice questions.

### 2.2. Questionnaire Design

The first section had four questions related to the participants’ demographics, namely age, marital status, education level, and place of residence in Saudi Arabia. The second section of the survey contained four multiple-choice questions regarding child characteristics, namely age of the child, gender, site of pain, and duration of pain. The third section contained multiple-choice questions regarding the non-pharmacological methods used by Saudi mothers at home for pain management. The fourth section of the survey related to the practice followed by mothers to manage their children’s pain and consisted of only one multiple-choice question, namely, “In what circumstances will you consult a physician to help with your child’s pain management?” The last section of the survey was intended to study the attitude of mothers towards pain medication. It consisted of a total of 15 questions to be answered on a 3-point Likert scale (agree/disagree/unsure) and was adopted from a previous study conducted by Fortier et al. [9].

The questionnaires were translated to the Arabic language using the assistance of an Arabic-speaking senior professor in the clinical pharmacy department and a certified Saudi Arabian translator. Before the survey questionnaires were distributed to the intended participants, a pilot study was conducted among a randomly selected group of 23 Saudi mothers. The pilot study was done to test the reliability of the questionnaires. The reliability was determined by using the Cronbach alpha value, which was found to be 0.81. The results of the pilot study were not included in the main study.

The validated Arabic questionnaires were used for data collection. Social media platforms were chosen as the potential medium for data collection. For the purpose of data collection, four female students were appointed and given the clarity about inclusion and exclusion criteria and the procedure of data collection through social media platforms. The data collators were strictly investigated by two senior academicians of the pharmacy college. The data collectors had to ensure that the questionnaires reached all the regions of Saudi Arabia. The questionnaires started with a pre-condition that stated that “Saudi females with at least one child are only eligible to fill in the questionnaire; females without children are not allowed to complete the questionnaires.”

### 2.3. Sample Size

According to previous reports, the prevalence of pain among pediatrics was 74.4% [13]. The sample size for the given study was calculated by using the following equation:*n* = z^2^ × *p* × q/d^2^
where *n* is the minimum sample size, z is the constant (1.96), *p* is the prevalence of pain (among pediatrics, it was 0.744%), q is (1 − *p*), Z is the standard normal deviation of 1.96 corresponding to the 95% confidence interval, and d is the desired degree of accuracy.
*n* = (1.96)^2^ × 0.744 (1 − 0.744)/(0.05)^2^*n* = 293

### 2.4. Data Management

The collected research data was carefully examined and extracted for any missing or incomplete responses, as it is an important step in performing a research study [14]. During this examination, we identified some incomplete questionnaires, as shown in Figure 1.

### 2.5. Data Analysis

The Statistical Package for Social Sciences (SPSS Inc., Chicago, IL, USA), version 25 was applied to analyze the data, and descriptive statistics, such as frequency and percentages, were applied. The Chi-square and Fisher’s exact tests were used to find any differences between the mothers’ demographic data and attitudes, at a significance level of 0.05.

## 3. Results

### 3.1. Study Population

A total of 399 mothers responded to the questionnaires. More than half of the participants (225; 56.4%) were aged between 25 and 35 years, while 119 (29.8%) were aged between 36 and 45 years. Nearly all participants were married (386; 96.7%). A large majority of the mothers (288; 72.2%) were university graduates, while 21 (5.3%) had studied only up to primary school. Detailed information on the demographic data is given in Table 1.

### 3.2. Child Characteristics

The number of boys and girls were almost equal; there were 200 (50.1%) girls and 199 (49.9%) boys. The mean age of the children was 4.9 (SD = 3.2). Some mothers reported more than one site of pain. The most reported site of child’s pain was mouth/throat (211; 52.9%), followed by abdomen (199; 49.9%), head (58; 14.5%), ears (69; 17.3%), and bones/joints (37; 9.3%). The frequency of the children’s pain, as stated by the mothers, was mostly less than once a month (196; 49.1%), followed by once in a month (137; 34.3%), and once a week (48; 12%) (Table 2). Differences in sites of pain and duration between the two genders (*p* < 0.05) are reported in Table 2.

The most commonly reported non-pharmacological methods used by mothers at home for the management of their children’s pain were the following: letting them take rest or sleep (250; 62.6%), feeding them with fluids (228; 57.1%), applying cold packs (161; 40.4%), providing massage therapy (147; 36.8%), using warm packs (141; 35.3%), and taking the child to play (119; 29.8%) (Table 3).

The study revealed that most of the mothers (198; 49.6%) consulted physicians to obtain medication prescriptions and treatment recommendations to relieve their child’s pain, while 144 mothers (36.1%) consulted physicians only to obtain medication prescriptions, and 34 mothers (8.5%) consulted pharmacists to obtain drugs. The study also revealed that the most commonly used pain medication amongst Saudi mothers was paracetamol (316; 79.2%). However, some of the surveyed mothers also used antibiotics (8.5%) and ibuprofen (7.8%) (Table 4).

### 3.3. Mothers’ Attitudes Towards Pain Medication

About 236 mothers (82%) reported that pain medications must be used for children as little as possible, because of their probable side effects. More than half of the mothers believed that analgesics have many side effects (232; 58.1%). About half of the mothers (202; 50.6%) were worried about the side effects of pain medications. Again, more than half of the mothers felt that pain medication works best when it is given as little as possible (210; 52.6%). A total of 164 mothers (41.1%) felt that children who take pain relief drugs to treat pain may get used to taking drugs to treat other illnesses as well, and 155 mothers (38.8%) felt that children will become dependent on pain medications if they start using them to treat pain. About 251 Saudi mothers (62.9%) felt that giving a child pain medication for pain would teach them about the proper use of medicines.

With respect to mothers’ negative attitudes and misconceptions towards pain medications, 98 mothers (24.6%) felt that analgesics are additive, and 117 mothers (29.3%) opined that the use of analgesics to treat children’s pain might lead them to start drug abuse later on in their lives. However, 64 (16%) of them felt that there is little need to worry about side effects of pain medication, while 117 (29.3%) of them felt that there is little risk of addiction when analgesics are given for pain. For more information about mothers’ attitudes towards pain mediation, refer to Table 5. In addition, this study revealed that there is some level of association between attitudes and the education levels of the mothers as well as the age of the mothers, as shown in the Appendix A.

## 4. Discussion

Management of pediatric pain is essential to avoid subsequent negative outcomes, such as the development of anxiety later on in life to face medical procedures, decreased pain perception and sensitization, and increased analgesic requirements [15]. Therefore, the concerned organizations, such as the American Pain Society (APS), and accreditation agencies, such as the Joint Commission on Accreditation of Healthcare Organizations (JCAHO), have released instructions and set standards to provide guidance on the appropriate pain-relieving methodologies for children [16]. In addition, the World Healthcare Organization (WHO) also provides a tool called *The WHO Analgesic Ladder*, which helps clinicians to select the most suitable pain relievers based on the patient’s condition [17]. However, all these efforts cannot be implemented effectively without the cooperation of mothers. This is because mothers are usually the primary caretakers of children, and so they are the ones who have to seek medical consultation for their children’s pain. Therefore, it is important that they have a positive perception and attitude towards taking medical assistance in dealing with pediatric pain.

One of the main barriers to appropriate pediatric pain management is the fear of adverse effects and addiction to analgesics [18]. This fear was majorly observed in our surveyed mothers, as was evident from the fact that 81.7% of them emphasized the scarce usage of analgesics to avoid adverse effects, 58.1% believed that pain medications have many adverse effects, 50.6% worried about the adverse effects, and 44.1% felt that children will become addicted to pain medication if they start taking it for pain. Other studies have reported similar results regarding belief in such myths and resultant negative attitude towards pain medication among parents of different ethnicities worldwide. Around 47% of White American parents, 69% of Hispanic parents, and 36% of British parents believe that their children should use analgesics as little as possible to safeguard themselves from probable adverse side effects. About 43% of Hispanic parents, 14% of White American parents, and 8% of British parents think that children will become addicted to pain medication if they take it for pain [9,18]. In regards to pain status, children with post-surgical pain were reported to have been inadequately managed by their parents because 52% and 73% of them were concerned about addiction and adverse effects, respectively [8].

Such fear of adverse effects might stimulate mothers to utilize non-pharmacological strategies to help their children cope with pain. Although published data about the various non-pharmacological strategies used by mothers around the world for home management of child pain is scarce, we were able to find commonalities between some of these strategies utilized by our participants and mothers from Western countries [6]. According to the American Academy of Pediatrics (AAP) and American Pain Society (APS), cognitive behavior strategies, such as imagery and relaxation, massage, and hot and cold compresses were among the most common non-pharmacological techniques used by mothers to help their children in managing pain. Similarly, our participants reported letting the child take rest and sleep, feeding fluids, providing massage therapy, and using cold or hot compresses as the most commonly utilized non-pharmacological techniques [18].

Another barrier to appropriate pediatric pain management is a lack of knowledge [18]. The belief that “the less often children receive analgesics, the better they work” was found in 55% of the surveyed mothers. These results seem to be consistent globally. About 37% of British parents, 35% of White American parents, and 49% of Hispanic parents reported the same belief [9,19]. The myth that “pain medication works best if saved for when the pain is quite bad” were noticed in 35.8% of our subjects: 51% of British parents, 53% of White American parents, and 71% of Hispanic parents. Inadequate pain management due to believing in the myth that “the less often children receive analgesics, the better they work” or that “pain medication works best if saved for when the pain is quite bad” can affect children physically and emotionally [20]. Research has revealed that 20% of pediatric pain could become chronic if it is undertreated [21].

The negative consequences of undertreatment of pediatric pain by parents can be overcome by providing an educational intervention for parents. A study of 284 parents who were taking care of their children’s post-surgical pediatric pain at home was conducted to assess the pain intensity, satisfaction, and usage of analgesics. The participants were divided into two groups; an educational interventional group (which was provided with adequate information about pain management and had frequent follow-up and reinforcement) and a usual care group (which was given only written and verbal information prior to discharge). In addition, the pain satisfaction, practices, and amounts of analgesics administration were changed for parents in the educational intervention group [22]. Such educational interventions and programs were found to be successful in aiding parents to provide better care for their children’s diseases and ailments. They were also observed to increase the parents’ compliance to prophylactic medications in case of pediatric sickle cell anemia, to improve the management of asthmatic children, and to enhance the parents’ knowledge of childhood immunizations [23,24,25].

Our study was conducted online with a limited number of mothers, who might not be representative of the whole society regarding the attitudes of parents towards pediatric pain management. Also, the grade of pain and access to the healthcare system, which may affect the results, were not assessed. However, the study provides further insight on an issue that had been studied among different populations and ethnicities and requires further clinical studies and proactive initiatives to empower better management of childhood health issues and illnesses.

## 5. Conclusions

Our study has proved that Saudi mothers have several misconceptions about pediatric pain management, which can affect their children’s quality of life. Implementing educational and awareness programs about pharmacological management of child pain could play an essential role in alleviating the impact of these misconceptions and improving pediatric pain management.

## Figures and Tables

**Figure 1 ijerph-18-00348-f001:**
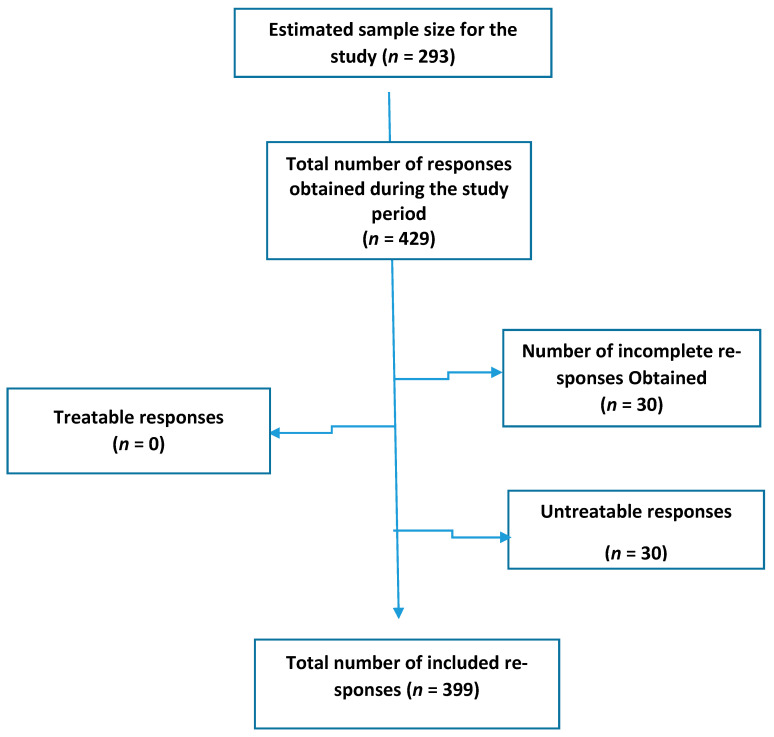
shows the data management.

**Table 1 ijerph-18-00348-t001:** Demographic data of respondents (*n* = 399).

Variables		Number (*n*)	Percentages (%)
Age of mothers	25–35 years 36–45 years 46–55years More than 55	225 119 52 3	56.4 29.8 13.0 0.8
Status	Married Divorced Widow	386 9 4	96.7 2.3 1.0
Education level	Primary/secondary school High school University Postgraduate	21 41 288 49	5.3 10.3 72.2 12.3
Region	Central Eastern Western North South	301 35 39 11 13	75.4 8.8 9.8 2.8 3.3

**Table 2 ijerph-18-00348-t002:** Characteristics of children and their pain.

Characteristics		Boys *n* (%)	Girls *n* (%)
Site of pain *	Abdomen ^¥^ Head Muscles Bones/joints Back ^¥^ Mouth Eyes Ears	88 (44.2) 25 (43.1) 15 (51.7) 19 (51.4) 2 (16.7) 110 (52.1) 7 (31.8) 29 (42.0)	111 (55.8) 33 (56.9) 14 (48.3) 18 (48.6) 10 (83.3) 101 (50.5) 15 (68.2) 40 (58.0)
Duration of pain	Less than once a month Once a month Once a week Several times a week Daily	196 137 48 17 1	49.1 34.3 12.0 4.3 0.3

* Participant can choose more than one answer. ^¥^ indicates significant difference between genders (*p* < 0.05).

**Table 3 ijerph-18-00348-t003:** Non-pharmacological methods used by mothers at home.

Non Pharmacotherapy	Boys *n* (%)	Girls *n* (%)
Taking the child to rest and sleep	111 (44.4)	139 (69.5)
Using cold pack	80 (49.7)	81 (50.3)
Massage	72 (49.0)	75 (51.0)
Training the child to relax	49 (48.0)	53 (52.0)
Applying Shower	53 (48.2)	57 (28.5)
Using warm packs	58 (41.1)	83 (58.9)
Using fluids	112 (49.1)	116 (50.9)
Take the child to play	57 (47.9)	62 (52.1)
Comfortable environment	52 (49.5)	53 (50.5)
Using natural products	28 (49.1)	29 (50.9)

**Table 4 ijerph-18-00348-t004:** Practices of mothers towards pain management.

Questionnaires	Number (*n*)	Percentages (%)
I consult the physicians only when giving medicine.	144	36.1
I consult a doctor to give the medicine to my child and for any methods to relieve the pain.	198	49.6
I don’t consult a doctor to give the medicine to my child or for any methods to relieve the pain.	23	5.8
I consult the pharmacists only when giving medicine.	34	8.5
Medications used: Paracetamol Ibuprofen Antibiotic	316 31 34	79.2 7.8 8.5

**Table 5 ijerph-18-00348-t005:** Mothers’ attitudes towards children’s pain medications.

Item NO	Questions	Disagree *n* (%)	Unsure *n* (%)	Agree *n* (%)
Q1	Children should be given pain medication as little as possible because of side effects.	22 (5.5)	51 (12.8)	236 (81.7)
Q2	Children who take pain medication for pain may learn to take drugs to solve other problems.	65 (16.3)	170 (42.6)	164 (41.1)
Q3	Pain medication works the same no matter how often it is used.	182 (45.6)	148 (37.1)	69 (17.3)
Q4	Pain medication works best when it is given as little as possible.	49 (12.3)	140 (35.1)	210 (52.6)
Q5	Pain medication has many side effects.	32 (8.0)	135 (33.8)	232 (58.1)
Q6	Children will be become addicted to pain medication if they take it for pain.	94 (23.6)	129 (32.3)	176 (44.1)
Q7	There is little need to worry about side effects from pain medication.	200 (50.1)	135 (33.8)	64 (16.0)
Q8	It is unlikely a child will become addicted to pain medication if taken for pain.	114 (28.6)	168 (42.1)	117 (29.3)
Q9	Pain medication is addictive.	142 (35.6)	159 (39.8)	98 (24.6)
Q10	Pain medication works best if saved for when the pain is quite bad.	126 (31.6)	130 (32.6)	143 (35.8)
Q11	Using pain medication for children’ pain leads to later drug abuse.	119 (29.8)	163 (40.9)	117 (29.3)
Q12	There is little risk of addiction when pain medication given for pain.	87 (21.8)	157 (39.3)	155 (38.8)
Q13	Side effects are something to worry about when giving children pain medication.	55 (13.8)	142 (35.6)	202 (50.6)
Q14	The less often children take pain medication for pain, the better the medicine works.	69 (17.3)	109 (27.3)	221 (55.4)

## Data Availability

The datasets analyzed during the current study are not publicly available, rather upon request from the corresponded author.

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
