# Peer review of "Assessment of Saudi Mothers’ Attitudes towards Their Children’s Pain and Its Management"

_ijerph, 2021, doi:10.3390/ijerph18010348_

Round 1

Reviewer 1 Report

I thank the editor for the possibility to revise the paper entitled “Assessment of Saudi Mother Attitudes Towards 2 Their Children Pain and its Management”. The study purpose was to evaluate the maternal attitude to manage the children’s pain.

Although the study was interesting, I think the paper should be revised consistently.

I think that the main aim of the paper was not clear. Why a descriptive analysis was important? What research field the paper extend? What clinical and research implication had the results?

Furthermore, I suggested several revisions before the publication.

Please, revise all text to correct typo. The English should be revised consistently by a native.

The references were poor and sufficiently recent, I suggested authors to extend and upload the literature.

The abstract was well organized and full of information.

The introduction section was readable.

Method section

Line 63: I think “what sup” is not the correct name of the social media app.

Was the study approved by an ethical committee?

Can the authors explain what “non-pharmacological methods” means?

Line 74: I think “licker” is not the correct name of the response method.

I think that it would be useful to report the items about the maternal attitude to manage the pain.

Cronbach alpha was usually reported as a decimal (.81) or percentage (81%) value.

Figure 1 should be improved both graphically and the content organization.  

The statistical analyses were not sufficient. For example, were comparative analyses between the socio-demo info carried out? Was the distribution of data normally? Etc etc

The section “Data analysis” should be improved. The authors could add more detailed information about.  

Results. More descriptive info were missing, e.g., mean and standard deviation of mother’ and child’s age.

Table 1 was not immediately clear, I suggested to format it adding internal line or space between section.

I suggested to include a section on the study limitation, clinical, and research implication of the study.

I hope to read te revised paper soon.

Author Response

Comments

Response

Line #

“I think that the main aim of the paper was not clear. Why a descriptive analysis was important? What research field the paper extend? What clinical and research implication had the results?”

Thank you for your valuable comments.

As we mentioned in the last paragraph, there is "scarce information about mothers’ attitudes towards pain management and analgesics use locally, and the aim of study is to assess the attitude of mothers toward their children pain and its management". Thus, the descriptive study using quantitative analysis is the most useful analysis since we don't know much about the topic locally. Furthermore, such descriptive studies can produce information that interests policy makers and rich data that lead to important recommendations.

“Please, revise all text to correct typo. The English should be revised consistently by a native.”

Thank you for your valuable comments.

The whole manuscript has been edited by certified agency. A copy of editing certificate is attached on the author menu.

“The references were poor and sufficiently recent, I suggested authors to extend and upload the literature.”

Thank you for your valuable comments

There are 25 references for the manuscript. Of which, around 8 references before 2005, and some of these are considered the still current practice of some clinicians relying on these references such as WHO & AAP/APS guidelines.

Line 63: I think “what sup” is not the correct name of the social media app.

Thank you for your valuable comments

WhatsApp is the world’s #1 messaging application and it's considered social media platform.

You can see more here https://unlimitedexposure.com/basic-digital-marketing/1097-whatsapp-is-a-growing-social-platform,-what-is-its-digital-marketing-potential.html .

A quotation from abovementioned website “WhatsApp has more than 1.6 billion monthly active users which is comparable to social platforms like Facebook and Instagram. More than 180 countries are represented in WhatsApp’s user base, and in 133 of these, it is the primary one-to-one messaging application."

Was the study approved by an ethical committee?

Thank you for your valuable comments

Yes, it does. It was part of big project and IRB will be attached on author menu.

Can the authors explain what “non-pharmacological methods” means?

Thank you for your valuable comments

We think it’s self-explanatory term and used frequently in the medical literature and textbooks. In addition, it’s clearly mentioned examples of these non-pharmacological methods in lines 122-125. Also, table 3 reports all the non-pharmacological methods used by participating mothers.  

Line 122-125

Table-3

Line 74: I think “licker” is not the correct name of the response method.

Thank you for your valuable comments

It was wrongly written. It’s corrected now to “Likert’s scale”

Line 78

I think that it would be useful to report the items about the maternal attitude to manage the pain.

Thank you for your valuable comments

We really don’t fully understand what the respected review aims to. If it’s something relating to typo or poor understanding, the whole manuscript has been edited by certified agency. A copy of editing certificate is attached on the author menu.

Cronbach alpha was usually reported as a decimal (.81) or percentage (81%) value.

Thank you for your valuable comments

It was wrongly written. It’s corrected now to “0.81”

Line 85

Figure 1 should be improved both graphically and the content organization.

Thank you for your valuable comments

We agree that figure needs improvement. We improved the figure to be easy to follow

Figure 1

The statistical analyses were not sufficient. For example, were comparative analyses between the socio-demo info carried out? Was the distribution of data normally? Etc etc

Thank you for your valuable comments

The comparative analysis was done, and more analysis were done.  

Table 2 , 4, 6

The section “Data analysis” should be improved. The authors could add more detailed information about.

Thank you for your valuable comments.

Chi square test was sued to find association between demographic data of mother  and attitude questions

Table 6

Results. More descriptive info were missing, e.g., mean and standard deviation of mother’ and child’s age.

Thank you for your valuable comments

The mean age of child was mentioned. While the mothers age was mentioned as categories 

Table 1 and 2

Table 1 was not immediately clear, I suggested to format it adding internal line or space between section.

Thank you for your valuable comments

It’s fixed to be more clear.

Table 1

I suggested to include a section on the study limitation, clinical, and research implication of the study.

Thank you for your valuable comments

It’s available in the last 2 paragraphs of the discussion part. In addition, some implication may be implied throughout the discussion.

Reviewer 2 Report

Thank you for submitting this manuscript, “Assessment of Saudi Mother Attitudes Towards Their Children Pain and its Management,” on a topic that needs more general public awareness. 

The author highlights a misconception and myths that may impact patient care and, if addressed, can improve the overall quality of life. I felt several areas needed some clarification or elaboration to improve the scientific accuracy and readability of the manuscript.

  1. The study design has an inherent limitation of “age group of the subjects from 1-12,” as it is challenging to locate the site of pain or discomfort in an infant and toddler who cannot verbalize, even for an experienced pediatrician sometimes.
  2. The author should clarify the inclusion and exclusion criteria used in their study.
  3. The author can elaborate and clarify in the manuscript why the child’s site of pain is exceeding more than 100%? I understand that the authors have mentioned the legends of Table 2. 
  4. There are a few typos and grammatical errors in the manuscripts, which the authors can recheck to correct them, like “use of worm pack” at few places. I assume that the author meant “use of warm packs.” 
  5. Figure 1 needs to be clarified, especially with the usage of terms “treatable and untreatable responses.” I am not sure what the author is trying to imply by the estimated sample size of 380, but the obtained responses were 429, and the selected ones were 399, which is more than the estimated sample size. Does this mean a single individual submitted multiple answers?
  6. Several factors are not considered, like the grade of pain and easy access to health care, which may lead to false generalization of the whole population. This should be acknowledged in the limitations.
  7. I agree with the general responses that analgesics usage is not without any side effects, and I agree with the author’s advocating judicious use of analgesics. So, the author should highlight/add the word “appropriate” analgesics usage rather than stressing on blanket usage of painkillers.
  8. The authors have appropriately concluded with a general interpretation of the results and provided possible future implications of this study, but the authors should elaborate more on their results, especially Table 3 in their discussion. 

Author Response

Comments

Response

Line #

The study design has an inherent limitation of “age group of the subjects from 1-12,” as it is challenging to locate the site of pain or discomfort in an infant and toddler who cannot verbalize, even for an experienced pediatrician sometimes.

Thank you for your valuable comments.

We totally agree on that pain of infant and pediatric below the age of 2 years can’t be expressed by the child; however, our aim was towards the mothers’ attitudes of children pain when THEY FEEL THERE IS A PAIN.    

The author should clarify the inclusion and exclusion criteria used in their study.

Thank you for your valuable comments.

It’s mentioned in the method section “Participants involved in the study were child bearing mothers of Saudi Arabia who are Arabic- speaking, and have children aged from 0 to 12 years”

63-64

The author can elaborate and clarify in the manuscript why the child’s site of pain is exceeding more than 100%? I understand that the authors have mentioned the legends of Table 2.

Thank you for your valuable comments

We agree, and thus, we added one sentence for clarification in the result section.

116-117

There are a few typos and grammatical errors in the manuscripts, which the authors can recheck to correct them, like “use of worm pack” at few places. I assume that the author meant “use of warm packs.”

Thank you for your valuable comments

The whole manuscript has been edited by certified agency. A copy of editing certificate is attached on the author menu.

Figure 1 needs to be clarified, especially with the usage of terms “treatable and untreatable responses.” I am not sure what the author is trying to imply by the estimated sample size of 380, but the obtained responses were 429, and the selected ones were 399, which is more than the estimated sample size. Does this mean a single individual submitted multiple answers?

Thank you for your valuable comments

We received 30 incomplete responses that are “untreatable” or in other word can’t be included in the study because it's missing the core information. Only one response from each mother. Figure 1 has been improved both graphically and the content organization to be understandable.   

Figure 1

Several factors are not considered, like the grade of pain and easy access to health care, which may lead to false generalization of the whole population. This should be acknowledged in the limitations.

Thank you for your valuable comments

We totally agree. These limitations such as grade of pain and access to healthcare system were added as limitation in addition to other limitations.

224-226

I agree with the general responses that analgesics usage is not without any side effects, and I agree with the author’s advocating judicious use of analgesics. So, the author should highlight/add the word “appropriate” analgesics usage rather than stressing on blanket usage of painkillers

Thank you for your valuable comments

We totally agree. It’s fixed now.

57

The authors have appropriately concluded with a general interpretation of the results and provided possible future implications of this study, but the authors should elaborate more on their results, especially Table 3 in their discussion.

Thank you for your valuable comments

We totally agree that we missed the discussion of non-pharmacological strategies of mothers for coping their children pain. A whole paragraph in the discussion is added now.

189-199

Round 2

Reviewer 1 Report

Dear authors, 

I appreciate your answers and clarification. 

The manuscript is significantly improved now. I approve the its publication.

Congratulations. 

kind regards

Author Response

I appreciate your answers and clarification. 

The manuscript is significantly improved now. I approve the its publication.

Congratulations. 

answer by author :

we are very much thank full to you and your team for giving a chance to correct our manuscript. We once again thanking you for all this efforts 

Reviewer 2 Report

Thank you for submitting this revised manuscript. The authors have done a decent job addressing the reviewer's comments, and the manuscript has significantly improved, but there are still minor areas that need some clarification.

  1. Indeed, the English writing style and grammar have improved, but there are still the same grammatical errors of "use of worm packs" at three instances, I pointed out in my previous review. I assume that the author meant "use of warm packs."
  2. Table 6 is too long, extending up to 4 pages, and the author should modify it for ease of reading or consider adding it as a supplemental table. 
  3. Can the author clarify as I still don't understand that the estimated sample size (380) is lower than obtained responses (429) when there is only one response for each mother? 

Author Response

Dear editor and team , firstly my heartfull thanks for your patience and reviewing our manuscript . We are glad to submit back our revision as suggested by reviewer 

  1. Indeed, the English writing style and grammar have improved, but there are still the same grammatical errors of "use of worm packs" at three instances, I pointed out in my previous review. I assume that the author meant "use of warm packs. "thank you for the comment and your time tp review , my sincear apologies for this error , i have corrected in the manusucript , you can clearly see in the whole manusucript 
  2. Table 6 is too long, extending up to 4 pages, and the author should modify it for ease of reading or consider adding it as a supplemental table. 

thank you for the comment and i have moved this table to supplemental table as you can see at he last of manuscript 

  1. Can the author clarify as I still don't understand that the estimated sample size (380) is lower than obtained responses (429) when there is only one response for each mother? 

my apologies for this confusion , actually we have collected the data for maximum number of saudi mothers and our estimated sample size was 293. 

According to previous reports the prevelance of pain among pediatrics was 74.4% [13]. The sample size for the given study was calculated by using the following the equation

N = z2 × p × q / d2

Where N is the minimum sample size, z is the constant (1.96), p is the prevalence of pain among pediatrics was (0.744%), q is (1-p), Z is the standard normal deviation of 1.96 corresponding to the 95% confidence interval, and d is the desired degree of accuracy.

N = (1.96)2 ×0.744 (1-0.744) / (0.05)2

N= 293

so we have distributed the questionnaires randomly, to reach maximum sample size to exclude bias during the study period over a 2 months, so we reached a total of 429 returned responses. However, during the data extraction and checking the data for accuracy and completeness of the questionnaires in the research, we found that about 30 questionnaires were incompletely answered by participants and therefore excluded from the study .so totally we included 399 responses which were completed by Saudi mothers, and those responses were greater than our estimated sample size and it is our study strength.

sample size was calculated by using the following the reference form Saudi Arabia .

  1. Elsayed AE, Alwosibei AA, Aldawsari AM, Alkhathlan MS, Al-Ghannam AK, Al-Sulaiman RM. Pain assessment and management for children hospitalized in the Pediatric Emergency Department, Military hospital, Al-Kharj, Saudi Arabia